# Knowledge of SARS-CoV-2 Epitopes and Population HLA Types Is Important in the Design of COVID-19 Vaccines

**DOI:** 10.3390/vaccines10101606

**Published:** 2022-09-24

**Authors:** Rafidah Lani, Nurul Aqidah Senin, Sazaly AbuBakar, Pouya Hassandarvish

**Affiliations:** 1Department of Medical Microbiology, Faculty of Medicine, Universiti Malaya, Kuala Lumpur 50603, Malaysia; 2Department of Biomedical Sciences, Faculty of Medicine, Universiti Malaya, Kuala Lumpur 50603, Malaysia; 3Tropical Infectious Diseases Research & Education Centre (TIDREC), Universiti Malaya, Kuala Lumpur 50603, Malaysia

**Keywords:** infectious diseases, SARS-CoV-2, COVID-19, vaccine, T cell epitopes, human leukocyte antigens

## Abstract

The COVID-19 pandemic has caused extensive loss of lives and economic hardship. In response, infectious disease experts and vaccine developers promptly responded by bringing forth candidate vaccines, some of which have been listed in the World Health Organization’s Emergency Use Listing. Notwithstanding the diverse worldwide population genetics, the vaccines thus far developed are generic in nature for use worldwide. Differences in the human leukocyte antigen (HLA) in different populations, variation of the T cell epitopes, and the propensity of SARS-CoV-2 genetic mutations left room for improvement of the vaccines. Here, we discussed the implications of COVID-19 vaccination and SARS-CoV-2 infection by taking into consideration SARS-CoV-2 mutations, T cell epitopes, risk factors, and current platforms of candidate vaccines based on the HLA types that are commonly present in Peninsular Malaysia Chinese, Indian, and Malay populations. The HLA types associated with protection against and susceptibility to severe SARS-CoV-2 infection were identified based on reported case-control and cohort studies. The relevance of including the non-spike SARS-CoV-2 proteins in the future COVID-19 vaccines is also highlighted. This review is meant to trigger researchers to acknowledge the importance of investigating the possible relationships between the HLA haplotype and the SARS-CoV-2 strains circulating in different populations.

## 1. Introduction

The COVID-19 pandemic has impacted our lives not only physically and mentally, but also economically. To date, approximately 580 million confirmed cases of COVID-19 and 6.4 million deaths have been recorded globally. The impact is no less devastating in Malaysia: with over 5000 confirmed cases daily, the total number of confirmed COVID-19 cases is approaching 5 million with almost 36,000 deaths [1].

The SARS-CoV-2 virus is the etiological agent responsible for several pneumonia-like cases that began in Wuhan, China [2]. Being the hub of transportation and industry for central China, the outbreak that started in early November, or December 2019, rapidly spread to become a pandemic [3]. Similar to other viruses that are transmitted through direct, indirect, or close contact with respiratory secretions or droplets from infected people [4], this *Betacoronavirus* spreading was greatly facilitated by international air travel [4]. The enveloped SARS-CoV-2 virus bears a large (approximately 30+ kb) single-stranded-positive sense RNA genome consisting of up to 14 open reading frames (ORFs) that are translated into the spike (S) protein, matrix (M) protein, envelope (E) protein, nucleocapsid (N) and about 16 non-structural proteins (nsps) [5]. Similar to other RNA viruses, SARS-CoV-2 also accumulates genomic mutations as it replicates owing to natural selections [6]. A number of mutations contribute to the augmented ability of the virus to replicate as well as to evade the host immune responses [6].

With the growing number of cases and the emergence of new SARS-CoV-2 mutants, infectious disease experts, epidemiologists, and public health officers have worked relentlessly to control the spread of the infection and at the same time to deduce the consequences of SARS-CoV-2 mutations. Just within a year since the COVID-19 pandemic started, vaccines have been manufactured and used by millions around the world. The exact mechanism of how SARS-CoV-2 caused severe COVID-19 disease, however, is still not known. Here, we look at the potential importance of the human leukocyte antigen (HLA) in COVID-19, Ref. [7] focusing on the multi-ethnic Malaysian population.

The Major Histocompatibility complex (MHC) system or HLA complex in humans is located on the short arm of chromosome 6 (6p21.3) [8]. Normally inherited as an en bloc from each parent in a no recombination event, linked HLA genes (HLA-A, -B, -C, -DR, -DQ, -DP) are combined as a HLA haplotype and transmitted on a single parental chromosome [9]. Abiding by its imperative functions in self-recognition, eliciting the immune response to an antigenic stimulus and to the regulation of cellular and humoral immunity, HLA class I antigens (HLA-A, -B, and -C) are expressed on the surface of all nucleated cells and platelets (except those of the central nervous system) [10] while the HLA class II antigens (HLA-DR, -DP, and -DQ) are expressed on antigen-presenting cells (APC) [10]. These highly polymorphic HLA loci are involved in antigen presentation to CD8+ T cells (HLA class I), natural killer cells, and CD4+ T cells (HLA class II) [11].

The fate of the SARS-CoV-2 virus and the outcomes of the infection are highly dependent on the efficiency of one’s immune system, particularly the T-cell immunity. Considering that the HLA haplotype occurs differently in different populations, the efficiency in SARS-CoV-2 viral clearance and disease progression in return are speculated to be varied. Studies associated with SARS-CoV-2 and HLA have focused on the involvement of cytotoxic CD8+ T and helper CD4+ T lymphocytes as their responses are vital for initial viral clearance, the development of immunologic memory, and eventually for orchestrating the adaptive immune responses [12]. In this report, we explored the potential repercussions of SARS-CoV-2 infection based on the HLA allele frequencies in the Malaysian population highlighting the HLAs that could contribute to the protection or exacerbation of SARS-CoV-2 infection.

### 1.1. SARS-CoV-2 Specific T Cell Epitopes

The search for potential vaccine targets has led to numerous studies to decipher the T cell epitopes that can evoke the MHC-I and MHC-II responses. In Table 1, we present the distribution of SARS-CoV-2 T cell epitopes as predicted from the combinations of the cohort (unexposed and convalescent individuals), bioinformatics, and mathematical modeling studies. Presentation of multiple SARS-CoV-2 epitopes is deemed critical in the induction of vaccine-based and natural infection immunity [13,14,15,16]. Detection of post-infectious T cell immunity is feasible through the employment of SARS-CoV-2-specific peptides even in seronegative convalescent individuals [13,17]. In the absence of antibody responses, specific T cell responses were observed in seronegative convalescent donors but not in unexposed donors, hence emphasizing the activation of T cell immunity upon infection. The SARS-CoV-2 CD4+ T cell is essential in evoking persistent and robust immune responses compared to the HLA class I T cell epitopes [13]. CD4+ T cell recognizes multiple dominant HLA-DR T cell epitopes [13]. The SARS-CoV-2 M protein was recognized by specific CD4+ T cells in COVID-19 cases [15]. The inadequacy of quality class II epitopes from the M protein is contributed to by its small size [18]. Although class II epitopes are predominantly available across the SARS-CoV-2 genomes, it appears that highly expressed proteins are preferred by memory CD4+ T cells [19].

T cell memory responses were observed more in patients recovering from severe COVID-19 infections than mild cases, with a greater magnitude of SARS-CoV-2-specific CD8+ T cells observed in the latter [20]. Detection of spike-specific antibodies in some donors with low (receptor-binding domain) RBD-specific antibodies indicates that antibodies were capable of targeting the non-RBD regions of the spike [20]. Establishments of T cell responses in correlation with milder cases will provide insight into protective immunity [20]. On that account, immense CD8+ T cell responses (to spike, M and N proteins) were observed in donors recovering from mild forms of the disease compared to severe cases [14,20].

These findings coincided with different subsets of SARS-CoV-2-specific T cells. Cytotoxic CD4+ T cells might not be a major contributor to SARS-CoV-2 clearance, since, unlike in influenza virus infections, CD107a+ CD4+ T cells (of cytotoxic potential) are scarcely detected [20]. Incorporating non-spike proteins such as N, M, and ORFs proteins in future vaccine design is perhaps beneficial as central memory and effector memory CD8+ T cells were identified in response to those proteins [20]. Ferretti et al., acknowledged in their study that next-generation vaccines incorporated with shared SARS-CoV-2 epitopes residing outside the spike protein will not only be independent of mutational variation but will also be better at eliciting SARS-CoV-2-specific CD8+ T cell immunity [16].

Heterologous immunity in SARS-CoV-2 infection is characterized by the pre-existing T cell responses against SARS-CoV-2 peptides [13]. The immunity is cross-reactive with common cold coronaviruses in 81% of unexposed individuals [13]. Mateus et al., 2020 also demonstrated the capability of SARS-CoV-2-specific memory CD4+ T cells to cross-react with corresponding ~67% homologous sequences from any of the many different commonly circulating common cold human coronaviruses (HCoV)-OC43, -229E, -NL63, and -HKU1 [19]. However, this event seems to happen uniquely in one direction and not vice versa [19]. Despite being highly speculative and vague, the implications of pre-existing HCOVs memory CD4+ T cells on the magnitude of SARS-CoV-2 infection are ascertained [19]. Although the magnitude of T cell responses is not associated with disease severity, severely ill patients possibly lack pre-existing SARS-CoV-2 T cells. This is demonstrated by lower recognition rates of SARS-CoV-2 T cell epitopes in individuals with more severe COVID-19 symptoms compared to non-hospitalized patients with high antibody titers [13].

While bioinformatics- and mathematical modeling-type studies have limitations of their own [21,22,23,24], cohort and case-control studies also come with some drawbacks [13,14,15,19,20,25]. The cohort and case-control studies discussed in this review are largely affected by the number of donors. Meticulous evaluation in comparing mild and severe cases is inconceivable without taking diverse T cell receptors, peptide-MHC affinities, and antigen sensitivities for different epitopes into consideration [13,14,15,19,20,25]. The significance of these factors is worthy of being addressed in future studies. Different techniques applied to determine IFN-γ-producing SARS-CoV-2-specific T cell responses yield contrasting results. This drawback is a result of detection method discrepancies as demonstrated by peptide-stimulated activation-induced marker (AIM) assays and ELISpots and ICS assays in a recent immunogenicity study of recombinant adenovirus type-5-vectored COVID-19 vaccine human phase I trial [14,26]. Although both methods are valid, the functional relevance is different.

The geographical regions where the studied donors are recruited also influenced cross-reactive responses as different coronaviruses (in both humans and animals) are circulating in different populations [27,28,29,30]. Pre-existing T cells exhibiting cross-reactivity do not necessarily imply previous coronavirus infections, but they could potentially be primed by other microbes too [31]. Thus, further detailed investigations based on this factor are necessary. Furthermore, the cohort studies focused on T cell responses in PBMCs instead of memory T cells at the site of infection most likely contributes to effective protection as observed in influenza virus infection [20].

### 1.2. Susceptible, Severe, and Protective HLA to SARS-CoV-2 Virus Based on Malaysian Population Allele Frequencies

The HLA class I and II loci identified in SARS-CoV-2 infection and epitopes recognition garnered from referred studies [15,19,20,21,22,25,32,33,34,35,36,37,38,39,40] based on the Malaysian population are presented in Figure 1 and Figure 2. We extracted the HLA phenotype frequency as shown in Figure 1 and Figure 2; from only “gold-standard” data sets comprising Malaysia Peninsular Chinese (n = 194) [32], Indian (n = 271) [33], and Malay (n = 951) [34] populations as available at allelefrequencies.net. The data set of HLA phenotype frequency in the Malaysian population is notably insufficient as it only covered HLA-A, -B, -C, -DRB1, and -DQB1 loci. This limited data set should be considered as an indicator for more investigations to be performed on Malaysian HLA phenotype frequency as it is vital not only for infectious diseases but for other diseases too. The HLA phenotype frequencies in these three main ethnicities in Malaysia are indisputably different to some extent. For example, HLA-DRB1*11:04 and -DQB1*04:01 are distinctly absent in Malaysia’s Indian population [33].

The HLA-DRB1*11:04 and -DRB1*03:01 are associated with protection against SARS-CoV-2 infection [35,36,37], while the latter is also associated with susceptibility and/or severe SARS and SARS-CoV-2 (when occurred as haplotype HLA-A*:01:01g-B*08:01g-C*07:01g-DRB1*03:01g) [35] infection in other studies together with HLA-DRB1*08:01, -DRB1*12:02, -DRB1*15:01 and -DQB1*06:02 [36,38]. Much as it is affected by the number of populations being studied (n), the phenotype frequency of HLA-DRB1*03:01, -DRB1*12:02, and -DRB1*15:01 are highest in Chinese, Malay, and Indian populations, respectively [32,33,34].

Out of the 35 most studied HLA class I phenotypes, six HLA phenotypes: HLA-A*02:01 (highest in Chinese) [32], -A*02:05, -B*15:03, -B*18:01 (highest in Malay) [34], -B*58:01 (highest in Chinese) [32], and -C*07:01 (highest in Malay) [34] are associated with protection [35,36,38] against SARS-CoV-2 infection and six phenotypes: HLA-A*01:01 (highest in Indian) [33], -B*07:03, -B*08:01, -B*27:07, -B*46:01 (highest in Chinese) [32] and -C*07:01 are associated with susceptibility and/or severe SARS-CoV-2 infection [35,38,39]. The HLA-C*07:01 is associated with both outcomes in studies with contradictory results. It has been reported that the haplotype ranked #1 HLA-A*01:01g-B*08:01-C*07:01g-DRB1*03:01g shows a significant correlation in both SARS-CoV-2 infection incidence and mortality. Contrariwise, haplotype ranked #2 HLA-A*02:01g-B*18:01g-C*07:01g-DRB1*11:04g shows a negative correlation suggestive to protection [35,36]. Subject to the number of populations being studied (n), all three populations are lacking in HLA-B*07:03 [32,33,34], while only the Chinese appear to have deficiency in HLA-A*02:05, -B*07:02, -B*18:01, -B*27:07, -B*44:02, and -C*02:02 [32]. Together with the Indian population, the Chinese population also lacked HLA-C*14:03 [32,33] and HLA-B*15:03 seems to be absent in the Malay population too [33,34].

The discrepancies between studies (case-control and/or cohort) pertaining to the association of HLA class I and II phenotypes with protection against; and susceptibility and/or severe SARS-CoV-2 infection are predicted to be greatly influenced by the population being studied and the alignment of the SARS-CoV-2 epitopes being used in the studies. For instance, HLA-DRB1*03:01 are associated with both protection against and susceptibility to SARS-CoV-2 infection in donors recruited from Oxford, the United Kingdom, and the Italian Bone Marrow Donor Registry (when occurred as haplotype HLA-A*:01:01g-B*08:01g-C*07:01g-DRB1*03:01g), respectively [20,35]. In addition, HLA-DRB1*03:01 is also associated with protection against SARS-CoV-1 infection in Taiwan’s healthcare workers [40].

The outcomes from studies revolving around HLA-C*07:01 point to the association of this particular phenotype to both protection against, and susceptibility toward severe SARS-CoV-2 infections [35,36]. When occurred as haplotype HLA-A*02:01g-B*18:01g-C*07:01g-DRB1*11:04g which is more frequent in the southern region of Italy, it is associated with protection against SARS-CoV-2 infection; and haplotype HLA-A*:01:01g-B*08:01g-C*07:01g-DRB1*03:01g which is more frequent in the northern region of Italy yields the opposite [35]. In another study involving a cohort in the Cagliari population (the southern region of Sardinia Island in Italy), the extended haplotype of HLA-C*07:01 is anticipated to provide protection against SARS-CoV-2 infection [36]. By scrutinizing all of the discussed studies, when it comes to investigating the relationship between HLAs and SARS-CoV-2 infection, it is desirable to begin research on SARS-CoV-2 sequences that are circulating in Malaysia, a bigger number of data set and diverse HLA phenotype in the Malaysian population; and all the more important is to identify the haplotype itself. From the two Italian studies, we can conclude that the outcome of SARS-CoV-2 infection is highly dependent on the polymorphism of particularly HLA-B*08:01, -DRB1*03:01, and -C*07:01 together with their combination with other alleles as a haplotype [35,36].

### 1.3. Host and Viral Factors in SARS-CoV-2 Infection

Among oft-mentioned analytical factors that are linked to SARS-CoV-2 infection, are host factors such as age [41,42,43,44,45], co-morbidities [46,47,48,49,50], and previous vaccinations [51,52,53] frequently brought up independently (if not investigated together with other confounding factors for example gender [54,55,56,57,58], demographical data [59,60,61,62,63], diet and exercise [64,65,66], past infection [67,68], food consumption [69,70,71], compliance with social-distance measure [72,73,74], etc.). A major concern is its mutation potentiality [75,76,77]. Age plays a substantial role as an independent risk factor for morbidity and mortality in COVID-19 patients [78,79,80,81,82,83]. This conception was proven in multivariate analyses in both Italy and China (despite their differences in mean age) where higher COVID-19 case-fatality rates were correlated with the increase in the median age [84,85,86]. This finding points to other indicators which are the differential expression of angiotensin-converting enzyme (ACE2) according to age since this enzyme is expressed in vital organs and its high expression in lung tissues of higher age groups is speculated to explain clinical pathology and susceptibility to SARS-CoV-2 infection [87,88].

In the case of ACE2, conditions such as hypertension, diabetes, cardiovascular disease, and chronic obstructive pulmonary disease (COPD) are also described as risk factors [89,90,91,92,93]. As a receptor for SARS-CoV-2 and its vast distribution in the lung, adipose, and endothelial tissues, COVID-19 disease progression is associated with the above-mentioned co-morbidities [94,95]. The usage of an angiotensin-converting enzyme inhibitor (ACEI) and angiotensin-receptor blocker (ARB) therapy is debatable in our early endeavor with COVID-19 infections since antihypertensive medications can modulate the expression of ACE2 protein [96]. With the exclusion of ACE2 polymorphism, the genetic association with hypertension traits (hopefully) involving larger cohorts in future studies, the safety of continuing the consumption of ACEI/ARB among patients is, for now, certain [96].

The impact of SARS-CoV-2 infection on diabetic patients is also being observed. The differential expression of ACE2 protein in normal and diabetic patients was reflected by the ACE2 expression in their lung tissue, liver, and pancreas. The expression of ACE2 is higher in bronchial and alveolar [97], pancreatic islets [98], and liver [99] of subjects with diabetes compared to normal subjects. The ramifications of the increase in ACE2 expression in these three organs of diabetic patients on SARS-CoV-2 infection and COVID-19 pathology warrants further investigation. As other diseases are being hypothesized to be co-morbidities and contributing factors to the exacerbation of COVID-19 clinical outcomes, COPD is also associated with the increase in mortality rates of COVID-19 patients [100]. Together with smoking behavior, COPD sets a greater risk for progression towards severe COVID-19 outcomes [101,102,103].

Although previous vaccinations especially the Bacille Calmette-Guérin (BCG) vaccine extensively featured in SARS-CoV-2-related studies, researchers failed to come out with sound evidence to propose and justify the usage of BCG for prevention of COVID-19 [104]. A number of new SARS-CoV-2 variants emerging as a result of the mutation have made the efficiency of recently developed vaccines questioned by the public. A theoretical study by Agerer et al., 2021 demonstrated that the nonsynonymous point mutations in SARS-CoV-2 MHC-I-restricted epitopes enable the virus to hide from CD8+ T cell surveillance [105]. Toyoshima et al., identified in their recent findings that ORF1ab 4715L and S protein 614G variants are strongly correlated with fatality rates [76]. However worrisome the impact of SARS-CoV-2 mutations on mortality rate is, the most controversial mutations such as the ones involving spike D614G, P323L and N501Y are only responsible for higher viral load and younger age of patients; enhancement of SARS-CoV-2 transmission capacity and improvement of viral fitness in different geographical regions [106,107]. All things considered, our previous remark on including the least-mutated, non-spike, T-cell epitopes in SARS-CoV-2 vaccine development still stands.

### 1.4. Current Vaccine Options for SARS-CoV-2 Virus

In response to the COVID-19 pandemic, vaccine developers raced towards getting a safe and efficient vaccine through various platforms. As of February 2021, 63 candidate vaccines (with two being suspended from development) are in clinical trial phases and 174 are in pre-clinical evaluations [108]. Among the candidate vaccines, about 32% or the plurality of the candidates are developed using protein subunit platform and the rest are non-replicating viral-vector (VVnr,16%), DNA (13%), inactivated virus (14%), RNA (11%), replicating viral-vector (VVr, 5%), virus-like particle (3%), VVr + antigen-presenting cell (3%), live-attenuated virus (2%) and VVnr + antigen-presenting cell (2%) [108]. 

Conventionally, two doses of vaccines (60% of the candidates) are required with a 14, 21, or 28 days interval in adherence to the prime-boost regimen, while 19% of the candidate vaccines only require one dose and 2% require three doses with the administration on day 0, 28 and 56 [108]. Most of the candidate vaccines are administered intramuscularly (76%), whereas 5% are intradermal and 3% are subcutaneous [108]. Currently, at the dawn of its pre-clinical development, two potential candidate vaccines are deemed to be convenient for vaccination as they are to be administered intranasally (Coroflu) [109] and as a skin patch (PittCoVac) [110], if successful. Recently, WHO has added five vaccines; BNT162 (Pfizer, USA), mRNA-1273 (Moderna, Cambridge, MA, USA), AZD-1222 (SK BIO, Seongnam, Korea), ChAdOx1_nCoV-19 (Serum Institute of India, Pune, India) and Ad26.COV2.S (Johnson & Johnson, New Brunswick, NJ, USA), into their Emergency Use Listing (EUL) for COVID-19 pandemic as the procedure, which has also served for the West Africa Ebola outbreak (2104–2016) [111]. More candidate vaccines that are listed in WHO EUL/PQ (prequalification) evaluation process are still waiting for a decision.

Similar to other countries, Malaysia started its National COVID-19 Immunisation Programme at the end of February 2021, immediately after acquiring five COVID-19 vaccines; BNT162 (Pfizer, New York, NY, USA), AZD-1222 (AstraZeneca, London, UK), CoronaVac (Sinovac, Beijing, China), Ad5-nCoV (CanSinoBIO, Tianjin, China), and Sputnik V (Gamaleya Research Institute of Epidemiology and Microbiology, Moscow, Russia) [112]. A total of 66.7 million doses were to cover 109.65% of those in the country in three phases [112]. Albeit short-lasting, mild to moderate COVID-19 vaccine side effects were anticipated and have been reported. Rare adverse events were also proclaimed however irrelevant the events were to the to COVID-19 vaccines. As of March 2021, 7 cases of blood clots in multiple blood vessels (disseminated intravascular coagulation, DIC) and 18 cases of cerebral venous sinus thrombosis (CVST) have been reviewed by European Medicines Agency (EMA)’s Pharmacovigilance Risk Assessment Committee (PRAC), out of 20 million vaccinated people in the United Kingdom and European Economic Area.113 However, a causal link between these events and AstraZeneca’s COVID-19 vaccine is still not proven and its benefits still outweigh the risks although these events deserve further investigation [113].

Contemplating our genetic make-up, the genetic risk factors that might contribute to severe COVID-19 disease and severe post-vaccination side effects should also be considered. For example, the hypertension traits in human chromosomes as well as Factor V Leiden which is the most common form of inherited thrombophilia with the highest occurrence of heterozygosity rate in Europe [114,115,116,117]. Before denouncing COVID-19 vaccines, it is wiser to consider all plausible explanations and reasons that originate from risk factors around us.

## 2. Conclusions

Considering most of the vaccines were developed with spike protein and this protein is prone to mutation, there is always room for improvement. By considering the SARS-CoV-2 T cell epitopes and the HLAs that are associated with protection against, and/or associated with susceptibility/severity to SARS-CoV-2 infection; the future COVID-19 vaccine design is expected to be developed based on the non-spike proteins such as NP, M, and ORFs. As highlighted in this perspective review, investigating the polymorphism of HLA-B*08:01, -DRB1*03:01, and -C*07:01 in a population is crucial as the outcomes of COVID-19 disease and vaccination are highly dependent on these alleles. Together with potential pre-existing cross-reactive T cell immunity to SARS-CoV-2, the performance of this future vaccine could provide better COVID-19 clinical outcomes and influence the herd immunity of all epidemiological models.

## Figures and Tables

**Figure 1 vaccines-10-01606-f001:**
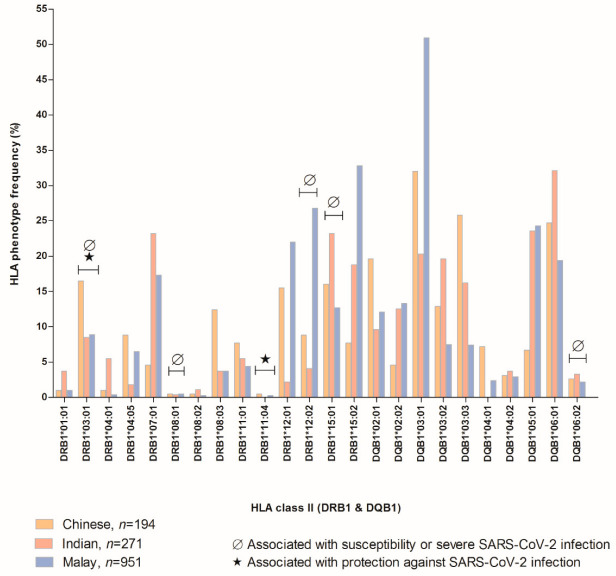
HLA class II phenotype frequency in Malaysia Peninsular Chinese, Indian, and Malay populations extracted from allelefrequencies.net in corresponding to SARS-CoV-2 epitopes recognition. The HLA phenotypes were collected from reported bioinformatics, case-control, and cohort studies.

**Figure 2 vaccines-10-01606-f002:**
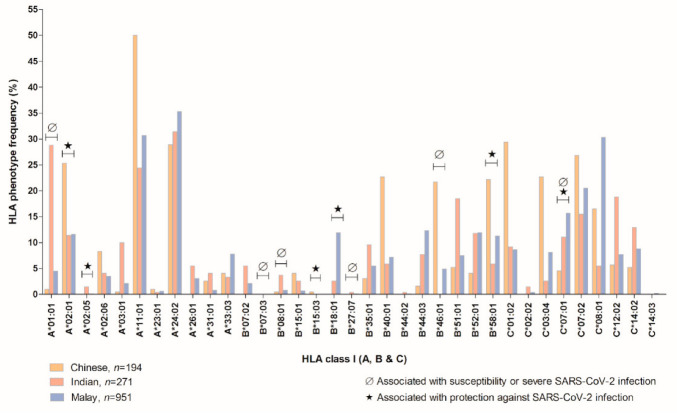
HLA class I phenotype frequency in Malaysia Peninsular Chinese, Indian, and Malay populations extracted from allelefrequencies.net in corresponding to SARS-CoV-2 epitopes recognition. The HLA phenotypes were collected from reported bioinformatics, case-control, and cohort studies.

**Table 1 vaccines-10-01606-t001:** Distribution of CD4+ and CD8+ epitopes based on SARS-CoV-2 proteins.

No.	Protein(s) and Their (Respective Numbers of Epitopes)	Subset	Ref.
1	N (4) non-RBD-S (90), RBD-S (23), E (2), ORF3a (7), ORF7a (3), ORF6 (7), ORF8 (4), nsp1 (1), nsp2 (7), nsp3 (11), nsp4 (10), nsp5 (4), nsp6 (9), nsp8 (2), nsp10 (2), nsp12 (9), nsp13 (4), nsp14 (4), nsp15 (1), nsp16 (2)	CD4	[19]
2	S (29), N (15), M (27), ORF7a (1)	CD4	[20]
3	S (11), N (21), ORF3a (4)	CD8	[20]
4	ORF9+N (50), ORF5+M (11), ORF2+S (3), ORF3 (4), ORF4+E (1), ORF6 (1), ORF8 (6)	CD4	[13]
5	ORF1 (4), ORF2+S (2), ORF9+N (5), ORF5+M (1)	CD8	[13]
6	S (1), ORF1ab (4)	CD8	[21]
7	M (2), S (5), ORF1ab (24), ORF3 (1), ORF6 (1), ORF7 (2), ORF8 (1)	CD4	[21]
8	nsp1 (2), nsp2 (14), PLpro (34), nsp4 (22), 3CL (6), nsp6 (18), nsp7 (4), nsp8 (4), nsp9 (2), nsp10 (2), RdRpol (19), Hel (14), nsp14 (19), nsp15 (4), nsp16 (9), S (20), ORF3a (10), E (8), M (8), ORF6 (6), ORF7a (4), ORF8 (3), N (7), ORF10 (1)	CD4	[22]
9	nsp1 (13), nsp2 (40), PLpro (128), nsp4 (40), 3CL (15), nsp6 (17), nsp7 (6), nsp8 (17), nsp9 (13), nsp10 (7), RdRpol (68), Hel (38), nsp14 (33), nsp15 (25), nsp16 (16), S (86), ORF3a (20), E (2), M (15), ORF6 (2), ORF7a (8), ORF8 (3), N (13), ORF10 (3)	CD8	[22]
10	N (8), nsp7 (1)	CD4	[15]
11	N (3)	CD8	[15]
12	ORF1ab (40), M (3), S (6), N (2), ORF3a (3), ORF7a (1)	CD8	[23]
13	ORF1ab (40), M (3), S (6), N (2), ORF3a (3), ORF7a (1)	CD4	[23]
14	ORF1ab (1478), S (248), ORF3a (69), E (18), M (72), ORF6 (8), ORF7a (26) ORF8 (22), N (60), ORF10 (12)	CD8	[24]
15	ORF1ab (1002), S (154), ORF3a (74), E (11), M (57), ORF6 (28), ORF7a (16) ORF8 (18), N (32), ORF10 (7)	CD4	[24]
16	M (5), N (2), S (5)	CD4	[25]
17	N (2)	CD8	[25]

CD4: cluster of differentiation 4; CD8: cluster of differentiation 8; N: nucleocapsid; S: spike; RBD-S: receptor-binding domain-spike; E: envelope; ORF: open reading frame; nsp: non-structural protein; M: matrix; PLpro: papain-like protease; RdRpol: RNA-dependent RNA polymerase; Hel: helicase.

## Data Availability

Not applicable.

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
