# Peer review of "Knowledge of SARS-CoV-2 Epitopes and Population HLA Types Is Important in the Design of COVID-19 Vaccines"

_vaccines, 2022, doi:10.3390/vaccines10101606_

Round 1
Reviewer 1 Report
I have reviewed this informative article. The quality of the abstract is not meeting the quality of scientific writing, which needs a high standard of writing to publish in outstanding journal.
Please revise the whole article and remove English grammar problems. I suggest the authors take English editing services from some agencies to improve the quality of this study. I am suggesting some studies. Please read these studies and improve your article.
Introduction section
I suggest that authors to read the suggested studies add the latest citations to the introduction, literature and method sections to enhance the quality of the study.
Su, Z., McDonnell, D., Cheshmehzangi, A., Li, X., & Cai, Y. (2021). The promise and perils of Unit 731 data to advance COVID-19 research. BMJ Glob Health, 6(5), e004772. doi:10.1136/bmjgh-2020-004772
Literature section:
Add literature section. You cannot delete this section. Read the suggested literature studies to enhance your work's quality. Add a few lines about studies on how education and social media can educate patients.
Yu, S., Draghici, A., Negulescu, O. H., & Ain, N. U. (2022). Social Media Application as a New Paradigm for Business Communication: The Role of COVID-19 Knowledge, Social Distancing, and Preventive Attitudes. Frontiers in Psychology, 13. doi:10.3389/fpsyg.2022.903082
Materials and Methods
This section is very weak. Please follow the suggested studies and improve your paper. The authors need to improve this section. I am recommending some good studies. Read the methods of these studies, and improve your paper. Suggested useful articles citations:
Result
Read the results of these studies, and improve your paper according to these studies in this section. Suggested useful articles citations
Zeidabadi, S., Mangolian Shahrbabaki, P., & Dehghan, M. (2022). The Effect of Foot Reflexology on the Quality of Sexual Life in Hemodialysis Patients: A Randomized Controlled Clinical Trial. Sexuality and Disability, 41(1), 1-12. doi:10.1007/s11195-022-09747-x
Discussion section:
Make a separate heading for the Discussion section. It should be around one page and a half. Improve the study and make it strong. See the recommended studies and improve your sections.
Conclusion
Highpoint creativity and scientific contribution of this study to the body of literature. The English level needs corrections to meet scientific merit for publication. I accept and endorse this manuscript for publication after minor corrections, as suggested.
Author Response
Point 1: I have reviewed this informative article. The quality of the abstract is not meeting the quality of scientific writing, which needs a high standard of writing to publish in outstanding journal.
Please revise the whole article and remove English grammar problems. I suggest the authors take English editing services from some agencies to improve the quality of this study. I am suggesting some studies. Please read these studies and improve your article.
Respond 1: Dear Reviewer, Thank you for your comments and suggestion. Our manuscript has been reviewed by two English expert persons and we have corrected those words and grammar mistakes.
Point 2: Introduction section
I suggest that authors to read the suggested studies add the latest citations to the introduction, literature and method sections to enhance the quality of the study.
Su, Z., McDonnell, D., Cheshmehzangi, A., Li, X., & Cai, Y. (2021). The promise and perils of Unit 731 data to advance COVID-19 research. BMJ Glob Health, 6(5), e004772. doi:10.1136/bmjgh-2020-004772
Respond 2: Dear Reviewer, Thank you for you suggestion. We used the most related and recent information according to our manuscript topic.
Point 3: Literature section:
Add literature section. You cannot delete this section. Read the suggested literature studies to enhance your work's quality. Add a few lines about studies on how education and social media can educate patients.
Yu, S., Draghici, A., Negulescu, O. H., & Ain, N. U. (2022). Social Media Application as a New Paradigm for Business Communication: The Role of COVID-19 Knowledge, Social Distancing, and Preventive Attitudes. Frontiers in Psychology, 13. doi:10.3389/fpsyg.2022.903082
Respond 3: Dear Reviewer, Thank you for suggestion. We are afraid the suggested article not related to the topic we are discussing in our manuscript and we cannot add it into our manuscript.
Point 4: Materials and Methods
This section is very weak. Please follow the suggested studies and improve your paper. The authors need to improve this section. I am recommending some good studies. Read the methods of these studies, and improve your paper. Suggested useful articles citations:
Respond 4: Dear Reviewer, this manuscript was submitted as a review paper and not as research paper. Therefore, we don’t have any methodology section in our submitted manuscript
Point 5: Result
Read the results of these studies, and improve your paper according to these studies in this section. Suggested useful articles citations
Zeidabadi, S., Mangolian Shahrbabaki, P., & Dehghan, M. (2022). The Effect of Foot Reflexology on the Quality of Sexual Life in Hemodialysis Patients: A Randomized Controlled Clinical Trial. Sexuality and Disability, 41(1), 1-12. doi:10.1007/s11195-022-09747-x
Respond 5: Dear Reviewer, The submitted manuscript is a review paper and data presented in the manuscript are collected and discussed from Gold-Standard data set. We have discussed all the extracted data accordingly with proper citation.
Point 6: Discussion section:
Make a separate heading for the Discussion section. It should be around one page and a half. Improve the study and make it strong. See the recommended studies and improve your sections.
Respond 6: Dear Reviewer, Thank you for your comment. As mentioned in our pervious respond, this is a review paper and we have discussed all the collected data from Gold-Standard data set.
Point 7: Conclusion
Highpoint creativity and scientific contribution of this study to the body of literature. The English level needs corrections to meet scientific merit for publication. I accept and endorse this manuscript for publication after minor corrections, as suggested.
Respond 7: Dear Reviewer, Thanks for the comment and suggestion. we have corrected our English errors in our manuscript
Reviewer 2 Report
Review comments on vaccines-1854021
Ms. Ref. No. vaccines-1854021
Title:Towards HLA-based COVID-19 vaccine for the Malaysian Population
Authors: Rafidah Lani, Nurul Aqidah Senin, Sazaly AbuBakar, Pouya Hassandarvish
Major comments:
In this review manuscript, authors tried to develop a strategy for designing vaccines based on the discussions aboutthe implications of COVID-19 vaccination and SARS-CoV-2 infection by taking into consideration of SARS-CoV-2 mutations, T cell epitopes, risk factors, and current platforms of candidate vaccines based on the HLA types that are commonly present inPeninsular Malaysia Chinese, Indian and Malay populations.However, I regret to say that the manuscript cannot be accepted for publication in “Vaccines” in the present form based on following reasons.
(1) This review paper aims to develop a new strategy for designing vaccines for Malaysian people as the title being indicated as “Towards HLA-based COVID-19 vaccine for the Malaysian Population”. Although authors described in Abstract as “we discussed the implications of COVID-19 vaccination and SARS-CoV-2 infection by taking into consideration of SARS-CoV-2 mutations, T cell epitopes, risk factors, and current platforms of candidate vaccines based on the HLA types that are commonly present in Peninsular Malaysia Chinese, Indian and Malay populations”, the presentation of the importance of HLA types in Malay populationsis not well organized in the manuscript. Further, this review paper lacks the deep consideration to develop a new COVID-19 vaccine for the Malaysian population.
(2) It seems that there is no special proposal for the development of vaccines considering theHLA types ofMalaysian populations.
(3) Although authors stated that “we can conclude that the outcome of SARS-CoV-2 infection is highly dependent on the polymorphism of particularly HLA-B*08:01, -DRB1*03:01, and -C*07:01 together with their combination with other alleles as a haplotype” (page 6, lines 222-224) based on two Italian studies, it is not clear how they can draw this conclusion.
(4) Similarly, authors stated in the conclusion section that “investigating the polymorphism of HLA-B*08:01, -DRB1*03:01, and -C*07:01 in a population is crucial as the outcomes of COVID-19 disease and vaccination are highly dependent on these alleles” (page 8 lines 325-327). Particularly, authors stated that HLA-C*07:01 showed contradictory results (page 6 line 193).
Minor comments:
There are many miss types; such as
(1) Page 1 line 33; “globally. 1” “1” might be a reference number?
(2) Page 7 lines 226-228; “Among oft-mentioned analytical factors that are linked to SARS-CoV-2 infection, are host factors such as age [41-45], co-morbidities [46-50], and previous vaccinations [51-53] are frequently brought up independently” Second “are” should be removed.
(3) Page 8 lines295-296ï¼›“has also served for the West Africa Ebola outbreak (2104-2016) [111].”  This part might be “which has also served for the Wesï½”Africa Ebola outbreak (2014-2016)”?
(4) Page 8, line 298; “Similar to that other countries” This part should be “Similar to other countries”?
(5) Page 8, line 306: “18 cases of (cerebral venous sinus thrombosis, CVST) have” This part should be “18 cases of cerebral venous sinus thrombosis (CVST) have”
(6) Page 8, line 320-325; These parts need revisions since two sentences are grammatically incorrect.
Author Response
Point 1: This review paper aims to develop a new strategy for designing vaccines for Malaysian people as the title being indicated as “Towards HLA-based COVID-19 vaccine for the Malaysian Population”. Although authors described in Abstract as “we discussed the implications of COVID-19 vaccination and SARS-CoV-2 infection by taking into consideration of SARS-CoV-2 mutations, T cell epitopes, risk factors, and current platforms of candidate vaccines based on the HLA types that are commonly present in Peninsular Malaysia Chinese, Indian and Malay populations”, the presentation of the importance of HLA types in Malay populationsis not well organized in the manuscript. Further, this review paper lacks the deep consideration to develop a new COVID-19 vaccine for the Malaysian population.
Respond 1: Dear reviewer, Thank you for the comment. This manuscript collects all the related information from Gold-Standard database (Table 1) and role of different SARS-CoV-2 protein in activation of Different CD cells base on the HLA phenotypes reported in Malaysian population. Line 102-120 we have explained the importance of designing new vaccine base on other SARS-CoV-2 protein in activation of different population of CD cells. And We also discussed the varieties of HLA in different Malaysian population and the importance of that in disease severity and mortality.
Point 2: It seems that there is no special proposal for the development of vaccines considering the HLA types of Malaysian populations.
Respond 2: Dear Reviewer thank you for your comment. We have already given our suggestions in conclusion regarding future vaccine development By taking into account the SARS-CoV-2 T cell epitopes and the HLAs that are associated with protection against, and/or associated with susceptibility/severity to SARS-CoV-2 infection; the future COVID-19 vaccine design is expected to be developed based on the non-spike proteins such as NP, M, and ORFs.
Point 3: Although authors stated that “we can conclude that the outcome of SARS-CoV-2 infection is highly dependent on the polymorphism of particularly HLA-B*08:01, -DRB1*03:01, and -C*07:01 together with their combination with other alleles as a haplotype” (page 6, lines 222-224) based on two Italian studies, it is not clear how they can draw this conclusion.
Respond 3: Dear Reviewer, thank you for your comment. The conclusion was made based on the two published papers which we have cited in our manuscript (ref 35 & 36). The two published studies have reported the correlation of the frequent HLA haplotypes in the Northern, Southern Italian population with both protection against, and susceptibility toward severe SARS-CoV-2 infections
Point 4: Similarly, authors stated in the conclusion section that “investigating the polymorphism of HLA-B*08:01, -DRB1*03:01, and -C*07:01 in a population is crucial as the outcomes of COVID-19 disease and vaccination are highly dependent on these alleles” (page 8 lines 325-327). Particularly, authors stated that HLA-C*07:01 showed contradictory results (page 6 line 193).
Respond 4: Dear reviewer, Thank you for your comment. I have added more information in our manuscript to make our statement clearer (line 193-197). "It has reported that haplotype ranked #1 HLA-A*01:01g-B*08:01-C*07:01g-DRB1*03:01g shows a significant correlation in both SARS-CoV-2 infection incidence and mortality. Contrariwise, haplotype ranked #2 HLA-A*02:01g-B*18:01g-C*07:01g-DRB1*11:04g shows negative correlation suggestive to protection."
Point 5: There are many miss types; such as
(1) Page 1 line 33; “globally. 1” “1” might be a reference number?
(2) Page 7 lines 226-228; “Among oft-mentioned analytical factors that are linked to SARS-CoV-2 infection, are host factors such as age [41-45], co-morbidities [46-50], and previous vaccinations [51-53] are frequently brought up independently” Second “are” should be removed.
(3) Page 8 lines295-296ï¼›“has also served for the West Africa Ebola outbreak (2104-2016) [111].”  This part might be “which has also served for the Wesï½”Africa Ebola outbreak (2014-2016)”?
(4) Page 8, line 298; “Similar to that other countries” This part should be “Similar to other countries”?
(5) Page 8, line 306: “18 cases of (cerebral venous sinus thrombosis, CVST) have” This part should be “18 cases of cerebral venous sinus thrombosis (CVST) have”
(6) Page 8, line 320-325; These parts need revisions since two sentences are grammatically incorrect.
Respond 5: Dear reviewer, thank you for noticing all the small mistakes. All the minor mistakes/errors have revised accordingly.

Reviewer 3 Report
This review discusses the impact of two vastly understudied aspects of the immune response to SARS-CoV-2: 1) variation in the human leukocyte antigen (HLA) types and their impact on the severity of disease; and, 2) the role of T cell epitopes, especially on non-spike proteins of the virus, in protection. First, the authors try to make the case that an understanding of precisely how variation in HLA haplotypes impacts the severity of disease in infected individuals constitutes an important, unexplored parameter. Secondly, given that the most commonly used vaccines for the virus are spike protein-based and that this protein undergoes a high rate of mutation, a strong case is made that more attention should be given to T cell epitopes on non-spike proteins such as M and N.
With respect to the former contention relative to HLA types, the authors make the case that the severity of disease is either positively or negatively affected by a particular haplotype. Unfortunately, however, they cite instances where a particular haplotype is associated with both high and low disease severity. Certainly, one or more of numerous other factors impacts disease severity. Thus, it seems that it will be difficult, if not impossible to causally link a particular haplotype to a particular disease outcome.
The second suggested avenue of investigation, T cell epitopes on non-spike viral proteins, seems much more amenable to study. As shown in Table 1, there are an abundance of candidate epitopes on multiple proteins, any of which could be highly protective. A strong case can be made, indeed, that incorporation of epitopes residing outside the receptor binding spike protein carries great potential to increase vaccine efficacy.
This being said, the challenge of establishing a direct link between an individual haplotype and disease severity is going to be difficult. The complexity of this challenge diminishes the importance of this review.
Author Response
Point 1: This review discusses the impact of two vastly understudied aspects of the immune response to SARS-CoV-2: 1) variation in the human leukocyte antigen (HLA) types and their impact on the severity of disease; and, 2) the role of T cell epitopes, especially on non-spike proteins of the virus, in protection. First, the authors try to make the case that an understanding of precisely how variation in HLA haplotypes impacts the severity of disease in infected individuals constitutes an important, unexplored parameter. Secondly, given that the most commonly used vaccines for the virus are spike protein-based and that this protein undergoes a high rate of mutation, a strong case is made that more attention should be given to T cell epitopes on non-spike proteins such as M and N.
With respect to the former contention relative to HLA types, the authors make the case that the severity of disease is either positively or negatively affected by a particular haplotype. Unfortunately, however, they cite instances where a particular haplotype is associated with both high and low disease severity. Certainly, one or more of numerous other factors impacts disease severity. Thus, it seems that it will be difficult, if not impossible to causally link a particular haplotype to a particular disease outcome.
The second suggested avenue of investigation, T cell epitopes on non-spike viral proteins, seems much more amenable to study. As shown in Table 1, there are an abundance of candidate epitopes on multiple proteins, any of which could be highly protective. A strong case can be made, indeed, that incorporation of epitopes residing outside the receptor binding spike protein carries great potential to increase vaccine efficacy.
This being said, the challenge of establishing a direct link between an individual haplotype and disease severity is going to be difficult. The complexity of this challenge diminishes the importance of this review.
Respond 1: Dear Reviewer thank you for your comment and your time to review our manuscript. To clarify with you It's the same allele but if exist in different haplotype combination that makes the outcome (severity or protection) different. It is stated clearly in the manuscript. And of course, numerous factors come together impacts disease severity. That's why other factors included in the manuscript
Round 2
Reviewer 2 Report
Review comments on vaccines-1854021-peer-review-v2
Ms. Ref. No. vaccines-1854021
Title: Towards HLA-based COVID-19 vaccine for the Malaysian Population
Authors: Rafidah Lani, Nurul Aqidah Senin, Sazaly AbuBakar, Pouya Hassandarvish
Major comments:
In the revised manuscript (vaccines-1854021-peer-review-v2), authors made appropriate revisions including minor ones (such as grammatical errors and etc..). Further, they made clear and appropriate answers to the comments raised by reviewers including me. Therefore, the revised manuscript can be accepted as a Review Paper for publication in “Vaccines” in the present form.

Author Response
Dear reviewer, Thank you for your time and all your useful comments to improve our manuscript. We appreciate your help and your time.
Reviewer 3 Report
This is an interesting review that raises some important issues that warrant further investigation due to the potential they offer for enhancing the efficacy of existing Covid vaccines and possibly development of new vaccine strategies. The authors make a strong case for the development of T cell epitopes on proteins other than the spike protein as vaccine targets. Also, they try to make the case for the association of specific haplotypes in a population with disease severity. However, while such associations may, indeed, exist, definitive causal relationships between a particular haplotype-disease severity combination will likely be more complex due to the plethora of other factors that affect disease severity. That being said, the authors have not overstated these findings and they have made a case for further exploration of this avenue of investigation.
Author Response
Dear reviewer, Thank you for your statement. We agreed with the reviewer and as the reviewer has stated we have not overstated these findings and they have made a case for further exploration of this avenue of investigation. That is the intention of this review at the first place, to give perspective to researcher to address from this point of view and define the causative relationship between those factors by embarking a research